# Study on the Evolution and Resilience of Rail Transit Time Networks—Evidence from China

**Rui Ding** †[ORCID], **Linyu Du** †, **Yiming Du** *, **Jun Fu, Yuqi Zhu, Yilin Zhang and Lina Peng**

College of Big Data Application and Economics (Guiyang College of Big Data Finance), Guizhou University of Finance and Economics, Guiyang 550025, China

* Correspondence: duyiming@mail.gufe.edu.cn
† These authors contributed equally to this work.

**Abstract:** In the network operation and management of rail transit systems, the occurrence of unexpected events causes damage to the network structure, further hindering regional accessibility performance and the function of the system. This study is based on the rail transit operation schedules in 2009, 2013, 2016, 2019 and 2022. We construct a directional weighted rail transit time network (RNNT) with train operation time as the weight, compare the betweenness centrality, sum of the shortest time path and entropy importance, etc., and quantitatively measure the network accessibility, connectivity and its resilience evolution. The results show that the current rail transportation network in China has a "small-world" effect, and there are a few stations with strong connections. The most densely distributed intervals of travel times between pairs of nodes changes from [440, 445] to [207, 210]. The fastest and best-performing disturbance to network connectivity and accessibility performance are both caused by the betweenness disturbance strategy. When the network connectivity remains 80% effective, the ratio of failed nodes under the static betweenness centrality strategy decreases from 3.96% in 2009 to 2.31% in 2022, with weaker connections between node pairs, and their network resilience diminishes. When the network accessibility remains 80% effective, the ratio of failed nodes under the static (dynamic) betweenness centrality strategy increases from 0.13% (0.13%) in 2009 to 0.20% (0.23%) in 2022. Therefore, the rail transit network can protect the corresponding rail stations based on the station ranking of the above strategies, and this research is beneficial to rail transit network protection and structure optimization.

**Keywords:** rail transit time network (RNNT); network resilience; network accessibility; network connectivity

## 1. Introduction and Literature Review

Due to its convenience, large volume and safety, rail transit has become one of the most mainstream transportation modes in the world. Many countries in the world have formed railway transportation networks of different modes, containing both high-speed railroads and general passenger railroads, and the structural evolution of the transportation network is closely related to regional characteristics. The United States has a long history of railway development, 80% of which is freight transportation. With the development of geo-economic changes, many countries in the world attach great importance to the role of railroads in strengthening regional economic, political, social and cultural ties and safeguarding national security, and regional road networks tend to be integrated. In the North American Free Trade Area, the United States, Canada and Mexico are closely linked by railway. In order to strengthen the connection between railroad routes in European countries, the construction of the pan-European railroad network has been gradually enhanced, and the construction of the pan-Asian railroad network is also being actively promoted in Asia. However, Japan Shinkansen is recognized as one of the safest high-speed railways in the world, and its operation safety management leads at the international level.



European railway lines have been extended to domestic countries to form a relatively stable network, and many urban rail transit networks have been extended in all directions of the cities. The "Outline of the Construction of a Strong Transportation Country" and "Outline of the National Comprehensive Three-dimensional Transportation Network Planning" issued by the Central Committee of the Communist Party of China (CPC) and the State Council have clearly proposed building an integrated urban transportation network and enhancing the resilience of China's rail transportation system in the future [1,2]. During the 13th Five-Year Plan period, China's high-speed railroad network has expanded to "eight vertical and eight horizontal" on the basis of "four vertical and four horizontal". The total scale of the railroad network reached 154,600 km in 2021, which accounted for 6% of the world's railroad mileage and is expected to reach 200,000 km by 2035. Further, the structural evolution of a transportation network is closely related to urban form [3]. Many countries regard railway construction as an important national policy for the development of transportation, and it is also a necessary trend for the construction and development of rail transit. Strategies to improve the service efficiency of a network and stabilize its structure have become an important element in the world railroad development. China has the most complex rail network of any country, with the most extensive coverage and the largest scale operation under construction in the world. More than 80% of major Chinese cities have near-high-speed railroads, and the accessibility between regions has been significantly improved [3]. High-speed railroad is therefore becoming a main theme of railway modernization in the world. Along with the development of metropolitan areas and city clusters, as well as meeting the objective requirements of establishing a comprehensive transportation system, railroads have become an important part of the rail transportation network.

While vigorously developing rail transit networks, railway transport systems are also faced with many risks and challenges: In 1998 in Germany, ICE1 high-speed train vehicle design errors led to derailment, and all ICE1 trains were shut down for inspection. In 2011, the Yong-wen line of China caused interruptions in regional travel for nearly 33 h, and the derailment of train T179 in 2020 interrupted the travel route for more than 21 h. In Japan, because of frequent earthquakes, there are many train derailment events.... The word "resilience" is derived from the Latin word "resiliere", which means to restore to the original state. Holling first introduced the theory of resilience into the field of scientific research and defined ecosystem resilience as a property that persists within a system. Defining the basis of resilience is the idea that a system can persist robustly or be destroyed suddenly [4]. The root causes of rail network failures are mainly due to infrastructure [5] and vehicle issues [6], and various types of natural disaster events, health and safety and other emergencies have imposed stricter requirements on the operation of rail transportation networks [7,8]. Therefore, it is important to strengthen the network risk analysis [9]. The network resilience analysis can applied to study the adverse effects of disruptions [10,11], preparing optimal infrastructure restoration and traffic recovery plans [12], and diverse disturbance considerations are beneficial for avoiding potential hidden threats [13]. Existing research argues that resilience can integrate the resilience and recovery capacity of transportation systems in facing external shocks [14], and the formation and evolution process of resilience will help us to understand the performance change patterns of rail transportation networks under environmental or manmade disturbances. This way, managers can clarify the identification of and metrics and optimization methods for network resilience, which is beneficial to the operation and emergency management of rail transportation network systems. In an environment in which urban connectivity is increasingly diverse and the transportation network structure is more complex, there are still shortcomings in the resilience of the rail network, and the ability of the network to cope with interference and to resist attacks needs to be enhanced.

Complex network theory originated in the early 18th century, and it was initially applied to graph theory and network topology in the field of mathematics. Watts et al. proposed the "small-world" network model, and Barabasi proposed the "scale-free" net-

work model, which opened up new perspectives in the study of complex networks, with an increasingly wide range of applications for complex networks. The network model can be used as a powerful tool for analyzing rail transportation networks. Scholars introduced static and dynamic topology metrics based on complex networks to evaluate the performance of transportation networks in terms of connectivity [15,16], accessibility [17], network structure characteristics, and resilience to destruction [18–20]. Regarding the study of rail transit networks, scholars have chosen quantitative analysis more often compared to qualitative analysis accounts. They eventually found that node disturbance strategies for systems are insufficient for resolving network failures, and network performance is more stable under random disturbance strategies. It is generally believed that more damaging nodes to the network are more important, and their disturbance can lead to failures much faster [21]. Wen used centrality to identify this node importance [22]. Disturbance strategies can also be considered in terms of node degree values, clustering coefficients [23] and other indicators. The literature has also considered transportation system operations in network resilience studies, including methods that weigh passenger flow data with connected edges [24] or that consider different situations such as roadway capacity and passenger emergency response [25], but these studies rarely involve the dimension of rail transit time. The railroad train running time and train arrangement are relatively fixed, and the important measure for maintaining the stability of the network is the resilience and recovery ability. Therefore, a method of constructing a directional time-weighted network model with the running time between stations as the weight value is more in line with realistic needs.

In summary, the existing transportation network research has achieved certain results, but there are fewer studies based on the evolution of the resilience of the rail network from the point view of rail transit time. These would help to further analyze structural performance changes, which indeed can help optimize rail line planning and emergency response strategies. The railroad network studied in this article is defined as an integrated transportation network of electric railway and non-electric railway, and our study is mainly based on rail transit routes on these integrated railroad networks. In this study, therefore, based on the railroad operation schedule for a total of five years, from 2009 to 2022, a directional time-weighted network model is constructed on the basis of a complex network. We then analyze the characteristics of the network, such as the small-world property, connectivity and accessibility. We further propose node importance evaluation indexes based on the degree, betweenness centrality and sum of the shortest paths, etc. Finally, we quantitatively compare the stable state of the network structure under different disturbance strategies and analyze network connectivity and accessibility. In sum, we measure the trend of network resilience evolution in order to provide a basis for railway transport system management to develop emergency response plans.

## 2. Network Structure Analysis

### 2.1. Construction of the Network Model

In this paper, we take China's rail transit stations as our research object and use Python to collect data from the China Railway official website, www.12306.cn (On 30 May 2013, 26 Decemeber 2016, 20 July 2019, 19 April 2022.), which contains the rail transit operation schedules in 2009, 2013, 2016, 2019 and 2022. These are filtered and cleaned with MATLAB tools, and the operation time between stations is used as the weight value to construct a directed rail transit time network (RTTN), which has an adjacency matrix of RT. Gephi 0.9.2 was chosen to analyze the node layout of the rail network.

The nodes in the network represent the train stations, and the two sides existing between the network nodes represent the running rail transit time between two stations. The network is a directed network, so the adjacency matrix is not a symmetric matrix, and the element values in the matrix represent the passage time from station i to station *j*. Here, 0 represents the absence of direct running trains between different stations. The network structure is a data set composed of an $N \times N$ adjacency matrix, which reflects the structural

characteristics of the rail transit network as well as the network function of the system. With $G$ representing the rail transit network, the corresponding mathematical model is constructed using graph theory methods as follows:

$$G = <V,E,W>, \tag{1}$$

$N$ is the number of nodes, and $V$ is the set of nodes.

$$V = \{v_i | i \in I \equiv \{1, 2, \ldots, N\}\} \tag{2}$$

$E$ is the number of trips running between the stations, connecting different nodes and denoted by $e_{ij}$.

$$E = \{e_{ij} = (v_i, v_j) | i, j \in I\} \tag{3}$$

where $W$ is the weight value of the edges, which in this case denotes the rail transit time between different stations; the total number of edges is denoted as $M$; and the adjacency matrix of the network is $A$.

$$A = \left[a_{ij}\right]_{N \times N} \tag{4}$$

Since the train has the property of running in a direction, there is no self-connection between nodes, so all self-connection is removed with = 0. The connection between nodes is defined as:

$$a_{ij} \begin{cases} 1, (v_i, v_j) \in V \\ 0, (v_i, v_j) \notin V \end{cases} \tag{5}$$

### 2.2. Network Structure Characteristics

With reference to scholars' research on complex networks [26–28], different parameters are chosen to analyze the characteristics of the RTTN, measure the operation of the transportation system and study the evolution of the RTTN.

### 2.2.1. Degree and Degree Distribution of Nodes

The number of edges of node $i$ connecting to other nodes is its degree value, which is the number of trains running at that station. A larger degree value indicates that the node is more important in the RTTN and reflects the local connectivity of the network structure.

The distribution function $p(k)$ can be used to represent the degree distribution of a network and measure the overall characteristics of the network. Here, $p(k)$ represents the probability that a random node has exactly $k$ connected edges, is the ratio of the number of nodes with degree $k$ to the total number of network nodes, and can determine the type of network structure. The random network degree distribution approximately obeys a Poisson distribution, and the degree distribution of many real-world networks, such as scale-free networks, conforms to a power-law distribution.

### 2.2.2. Clustering Coefficient

The clustering coefficient is used to describe the degree of aggregation of nodes in the RT network. $K_i$ is the degree of node $i$, and $E_i$ is the actual number of edges generated between neighboring nodes of node $i$. The clustering coefficient of the entire RT network is noted as $C$, which is the average of the clustering coefficients of all nodes in the network. $C$ represents the connectivity relationship between neighboring nodes from a certain arbitrary node. It reflects the degree of node aggregation, the connectivity of the overall network structure [29], and in real networks, nodes always aggregate with each other, so $0 < C < 1$:

$$C_i = \frac{2E_i}{K_i(K_i - 1)} \tag{6}$$

$$C_i = \frac{\sum_{i=1}^{N} C_i}{N} \tag{7}$$

### 2.2.3. Betweenness Centrality

In network analysis, centrality is generally used as an indicator to confirm the relative importance of a node or edge, and degree centrality and eigenvector centrality can only reflect the local importance of a station. The betweenness of rail network structure reflects the role played by and influence of rail stations in the network structure from a global perspective, so the betweenness centrality of complex networks is chosen to assist in indicating the importance of stations. It considers the proportion of all the shortest paths in the network. The issue of time cost makes people prefer routes with shorter travel times, and the stations with more shortest paths will increase in importance in the transportation network.

The betweenness of station $i$ is the ratio of the number of paths passing through station $i$ to the number of all the shortest paths among those between stations in the network. If the shortest paths between many node pairs pass through node $i$, then the node has a heavy weight in the network and belongs to the intermediary or indirect connection in the node pair.

$$BC_i = \frac{2}{(N-1)(N-2)} \sum_{s \neq t} \frac{n_{st}^i}{g_{st}} \tag{8}$$

$g_{st}$ is the number of shortest paths between station $s$ and station $t$, and $n_{st}^i$ is the number of shortest paths between station $s$ and $t$ passing through station $i$. In railroad rail transportation, stations with high betweenness centrality play a key role in the transit of passenger flow, reflecting the role of stations as bridges in the connection of other railroad stations [30].

### 2.2.4. Sum of Shortest Time Path

The shortest path length $d_{ij}$ is the shortest time path from node $i$ to node $j$. The shortest arrival time between different stations is represented in the rail network structure, and once an unexpected event occurs, passengers have other alternative paths. The sum of the shortest path $SD$ is the sum of the time required from node $i$ to all other nodes:

$$SD = sum\big(d_{ij}(min)\big) \tag{9}$$

### 2.2.5. Entropy Importance

It is generally believed that the entropy importance can deeply reflect the utility value of indicator information, and its basic idea is to determine the objective weight based on the magnitude of the variability of the indicator. The smaller the entropy value of a utility indicator, the greater the degree of variability of the indicator and the greater the contribution to the importance of the station, and thus the greater the corresponding weight [23]. In this study, the node degree value, clustering coefficient, betweenness centrality and the sum of shortest time path were selected to find the entropy importance, and the steps of the entropy method to solve the indicator weights were as follows, with n indicators, m evaluation indicators, and data normalization process $x_{ij}$ denoting the elements of the $i$-th row and $j$-th column of the matrix; the contribution of the $i$-th sample indicator under the $j$-th indicator was calculated as $P$.

$$P_{ij} = \frac{x_{ij}}{\sum_1^n x_{ij}} (j = 1, 2, \ldots, m) \tag{10}$$

Calculate the entropy value $e$ of the $j$-th indicator.

$$e_j = -k \times \sum_1^n p_{ij} \times log(p_{ij}), \ k = \frac{1}{ln(n)} \tag{11}$$

Calculate the coefficient of variation $g$ for the $j$-th indicator.

$$g_j = 1 - e_j \tag{12}$$

Calculate the weight $w$ of the $j$-th indicator.

$$w_j = \frac{g_j}{\sum_1^m g_j} \tag{13}$$

### 2.3. Network Accessibility

Accessibility is the spatial distance of any spatial point to other points within a certain territory, and it can better reflect the connection characteristics of space. In this study, the rail transit time accessibility model is established with travel time instead of spatial distance, and it does not include the waiting time and transfer time, considering only the vehicle running time.

$$T_{ij} = W_{ij}(min) \tag{14}$$

$T_{ij}$ is the accessibility node $i$ and node $j$. $W_{ij}(min)$ is the minimum travel time between two stations.

### 2.4. Network Resilience Analysis

In this study, the effects of realistic emergencies on the rail transit network lines can be considered a direct disturbance, that is, a direct disturbance to stations and inter-station running trips, and the disturbance strategy is constructed based on indicators of different network structure characteristics (see Table 1). Detecting the different changes produced by the rail transit time network when the disturbance occurs is of great theoretical and practical significance for analyzing the transportation network and operation process. The change in network performance can be measured by removing the stations in the network through the simulation algorithm in MATLAB software and then detecting the resilience of the RTTN. Among the corresponding disturbance strategies selected in this study, the node degree values, betweenness centrality, sum of shortest time path, clustering coefficients and the entropy importance are mainly determined.

**Table 1.** Summary of disturbance strategy methods.

| Strategy Number | Strategy Name |
|---|---|
| Disturbance strategy 1 | Degree value-based disturbance strategy |
| Disturbance strategy 2 | Random nodes-based disturbance strategy |
| Disturbance strategy 3 | Random edge-based disturbance strategy |
| Disturbance strategy 4 | Static betweenness centrality-based disturbance strategy |
| Disturbance strategy 5 | Dynamic betweenness centrality-based disturbance strategy |
| Disturbance strategy 6 | Clustering coefficient-based disturbance strategy |
| Disturbance strategy 7 | Sum of shortest time paths-based disturbance strategy |
| Disturbance strategy 8 | Entropy importance-based disturbance strategy |

The procedure of network disturbance is as follows: First, according to the structural characteristics and corresponding indicators of the measured initial state network, the ranking of different stations is determined using the corresponding strategy, and the affected station or edge is determined from the network in a certain order. Then, the station is removed from the RTTN, the characteristics and corresponding indicators of the corresponding network structure are measured again, and the percentage of change is obtained as the change of network resilience. The betweenness centrality is subdivided into static and dynamic: dynamic means that each node is reordered after each disturbance process, and then the node with the largest betweenness number is selected to continue the disturbance. However, the static disturbance strategy is only affected according to the ordering of the initial state. The random disturbance strategy is randomly sorted for each

station, and a random value is generated after each disturbance until the end of the process. The above process is carried out 100 times, and the average value of these 100 times is taken as the final result of the random disturbance strategy.

The resilience metrics covered in this study are developed from the two perspectives of RTTN connectivity and accessibility, denoted by the *D*-value and the *S*-value, respectively.

$$D = \frac{degree''}{degree} \tag{15}$$

Here, *degree* is the sum of RTTN node *degree* values in the original state, and *degree''* is the sum of network node degree values after the disturbance process.

$$S = \frac{path''}{path} \tag{16}$$

Here, *path* is the sum of the shortest distances between all node pairs in the original state of the RTTN, and *path''* is the sum of the shortest distance between all node pairs after the disturbance process.

## 3. Research Results

### 3.1. China's Rail Transit Network Structure Is Becoming More Solid

Using Gephi to analyze the corresponding network structure characteristics of the RTTN, the rail network connect relationship table was imported into Gephi, and the corresponding layout as obtained using the Fruchterman–Reingold algorithm (see Figure 1). It can clearly be seen that the number of important stations increases significantly and the influence range expands, and the connections between nodes increase and gradually gather, indicating that the association of RTTN stations is strengthening year by year and the network structure is more solid.

The degree refers to the number of stations connected to other stations. According to the statistics, the degree value of Shenyang North Station was the largest at 71 in 2013, and the degree value of Zhengzhou Station was the largest during the other four years. Zhengzhou is known as the "heart of China's railroad"; Zhengzhou Station is the only star-shaped high-speed railway hub in China, linking with other provincial capitals. The degree value of most of the stations is relatively small, whereas the stations with a larger degree prefer to connect with nodes with larger degrees, which reflects the "Matthew effect" of the rail network. The concentration of these hub stations provides more choices for passengers. The proportion of stations (with a degree value equal to 4) is the largest in each study period, with 46.1%, 40.6%, 26.7%, 30.5% and 28% in each period. Meanwhile, the proportion of stations with a degree value higher than 2 is 94.3%, 94.4%, 90.8%, 95.4% and 94.5% in each period. It is generally believed that in a directional network, nodes with degree values greater than 2 have the property of interchange stations, and they are related to the operational efficiency of other rail transit lines and influence each other. From 2009 to 2022, the total number of stations in the rail transit system shows a trend of first decreasing and then increasing, and the average value of degree values increases year by year, indicating that global planning measures are being implemented and the operation mode of the rail transit network system is more reasonable, and the importance of stations within the RTTN in China continues to increase.

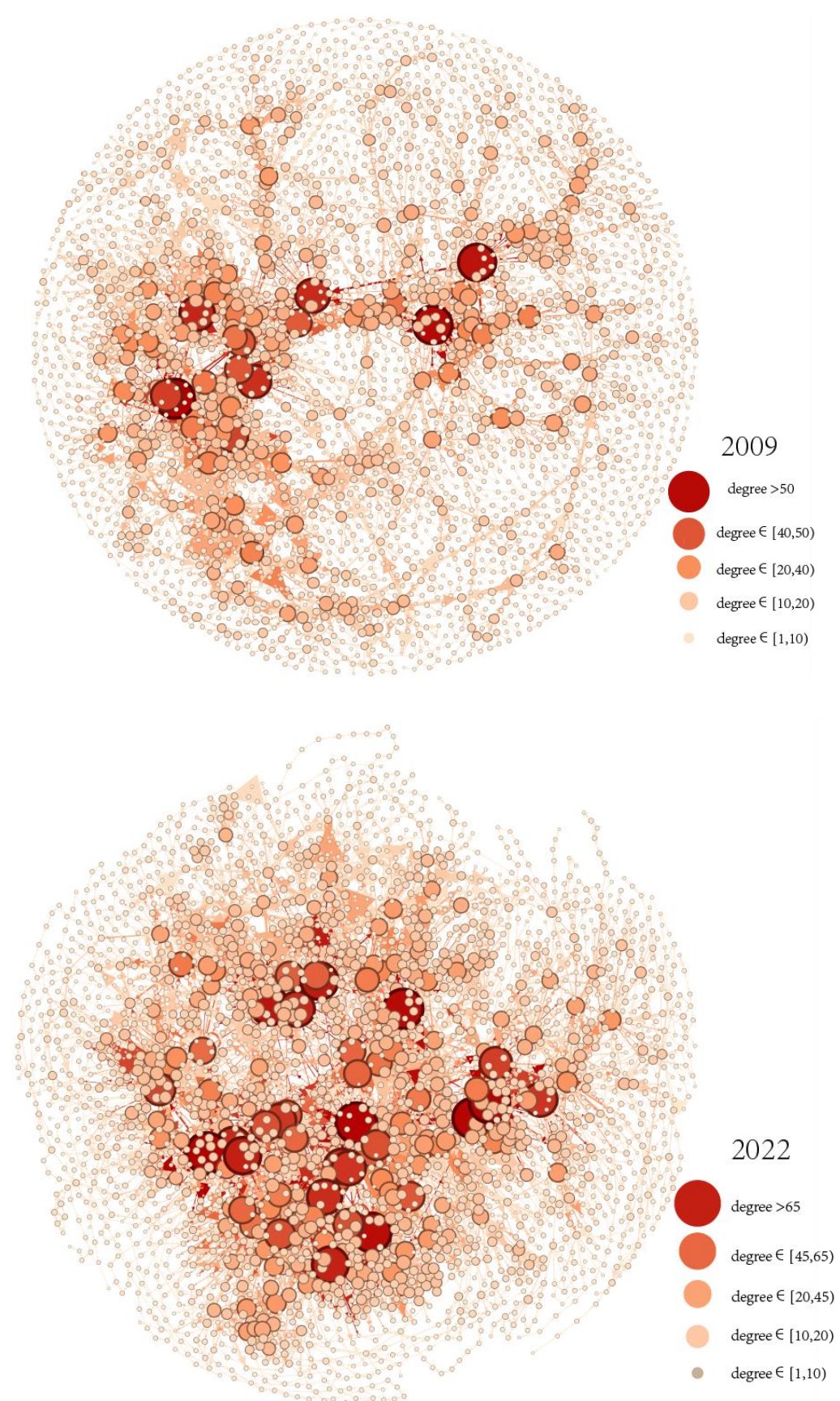

**Figure 1.** Railway station network layout in 2009 and 2022.

The visual analysis described above shows high-frequency key stations with directional connections between different stations. In order to analyze the structural characteristics of the rail network in more depth, the network structural indicators, such as the node degree values, clustering coefficients, average shortest path and modularity, will be analyzed (see Table 2). The average degree rises from 3.19 in 2009 to 4.653 in 2022, and the whole network shows more efficient cooperation. The graph density indicates the current coverage of the network and the tightness of the nodes' connections [24], and its current value is very small at 0.001. The performance of the transfer between stations is not strong, as most station pairs do not yet have convenient access, and only a few key stations are tightly docked. However, it can still be seen that there is a numerical improvement in the closeness of the rail network in 2022 compared to previous years. In this study, random network graphs of the same size for each year were simulated, and the average path length of the random graphs was larger, whereas the average shortest path of the rail transit network was smaller, based on the same average clustering coefficients, indicating that the RTTN exhibits a small-world effect. The small-world model was proposed and introduced by Watts and Strogatz in 1998 for the study of complex networks, which can be determined in terms of both a higher agglomeration coefficient and a shorter average path length. Most networks in real social activities have a small-world effect. This means that most nodes are only closely connected to their neighbors, which can explain the emergence of multiple network forms and facilitate the establishment of close cooperation within the network. Any other node in the network can be reached by passing through only a few nodes, and random reconnections occur between some pairs of distant nodes, which builds bridges between groups across small groups with strong internal relationships. Despite the low network density, the average path length values gradually decrease from 2009–2022, so the closeness is increasing in the whole rail network. The small-world effect allows stations in the network to have more opportunities to connect with other stations, and such an operational network is of greater potential. These stations form various small networks of rail clusters around different core nodes, but they perform poorly in terms of station connectivity on a larger scale. We see that this situation is mitigated in 2022 with increased inter-regional rail traffic connections and improved overall performance.

**Table 2.** Railway network structure characteristics indicators.

| Year | Number of Nodes | Number of Edges | Average Degree Value | Average Weighted Degree | Graph Density | Average Path Length | Average Clustering Coefficient |
|------|-----------------|-----------------|----------------------|-------------------------|---------------|---------------------|--------------------------------|
| 2009 | 3030 | 9666 | 3.190 | 553.25 | 0.001 | 11.708 | 0.265 |
| 2013 | 2783 | 9816 | 3.528 | 722.84 | 0.001 | 11.048 | 0.319 |
| 2016 | 2740 | 10,673 | 3.904 | 887.89 | 0.001 | 9.966 | 0.367 |
| 2019 | 2998 | 13,160 | 4.390 | 1057.82 | 0.001 | 9.000 | 0.411 |
| 2022 | 3068 | 12,122 | 4.653 | 987.34 | 0.002 | 7.874 | 0.397 |

The degree values and distributions are measured for each year, as shown in Figure 2. The horizontal coordinate is the result of the logarithmic operation of the degree value, and the vertical coordinate is the result of the operation of the degree distribution corresponding to the degree value. Using the least squares estimation to fit a straight line of linear distribution function, the degree distribution of each ten-year railroad traffic network can be represented by a linear function fit. It can be found that the absolute value of the slope of the degree distribution gradually decreases, that is, the non-homogeneity of the railroad network system weakens, the degree of the "core-edge" structure gradually decreases, and the development trend is positive.

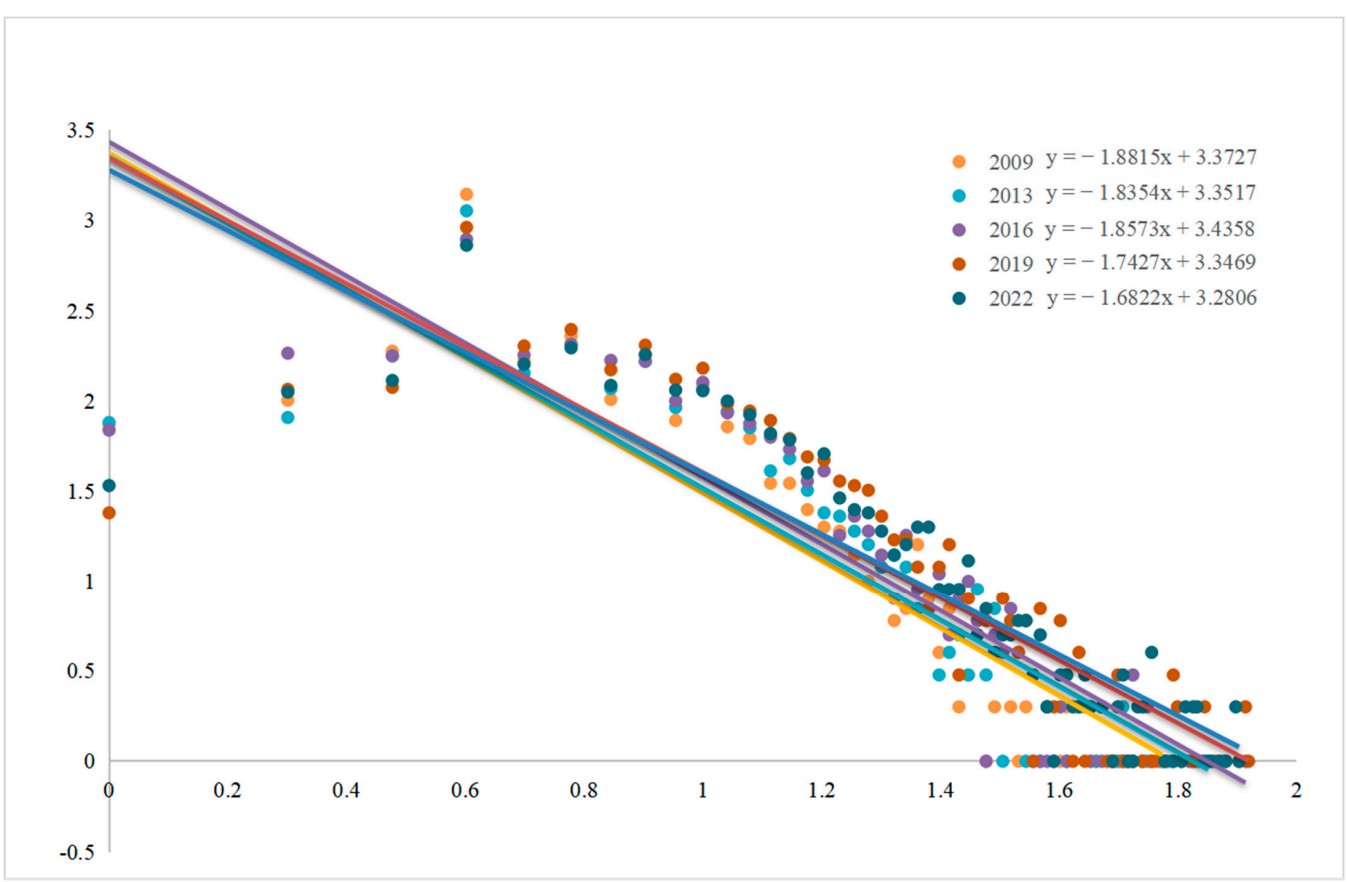

**Figure 2.** Double logarithmic scatter distribution of degree values and degree distribution.

### *3.2. Rail Network Accessibility Is Increasing*

In this study, the distribution of the running time between stations in each year is used to describe the accessibility of the rail transit network, and the length of the running time indicates the ease of access between nodes. In 2009, 66,254 pairs of node travel times were in the interval [440, 445], and the most densely distributed intervals in the other years were [505, 510], [405, 410], [325, 330], and [207, 210]. The values of the distribution intervals show an overall decreasing trend, and the values of the tails in the Poisson distribution of the values decrease year by year and tend to be stable. The distribution of rail transit time between 2019 and 2022 has the largest change; the optimization of the system and the improvement of the number of running trains are important reasons. Regional control caused by COVID-19 epidemic outbreaks in various regions and the stopping or rerouting of high-speed trains causes an overall decreasing trend in the transit time of the rail network. Shorter transit times can improve inter-regional connectivity and enhance the accessibility of the system network. The establishment of a regional rail network favors short-distance travel, and the development of a high-quality and diverse transportation network can significantly enhance the demand for medium- and long-distance travel, further encouraging more optimization of the transportation network by the rail service system.

### *3.3. Rail Network Resilience Evaluation*

3.3.1. The Resilience of Rail Network Connectivity Is Weakening Year by Year

The change in resilience of the connectivity of the rail network under each disturbance strategy is shown in Figure 3, and the connectivity performance validity of the network in the default initial state is 100%. After the failure of any node by disturbance, all edges associated with the node are destroyed. With the successive failures of important nodes, the impact on the network structure intensifies, the degree of association between nodes

weakens, the sum of degree values decreases, and the overall connectivity performance of the rail transit network system weakens.

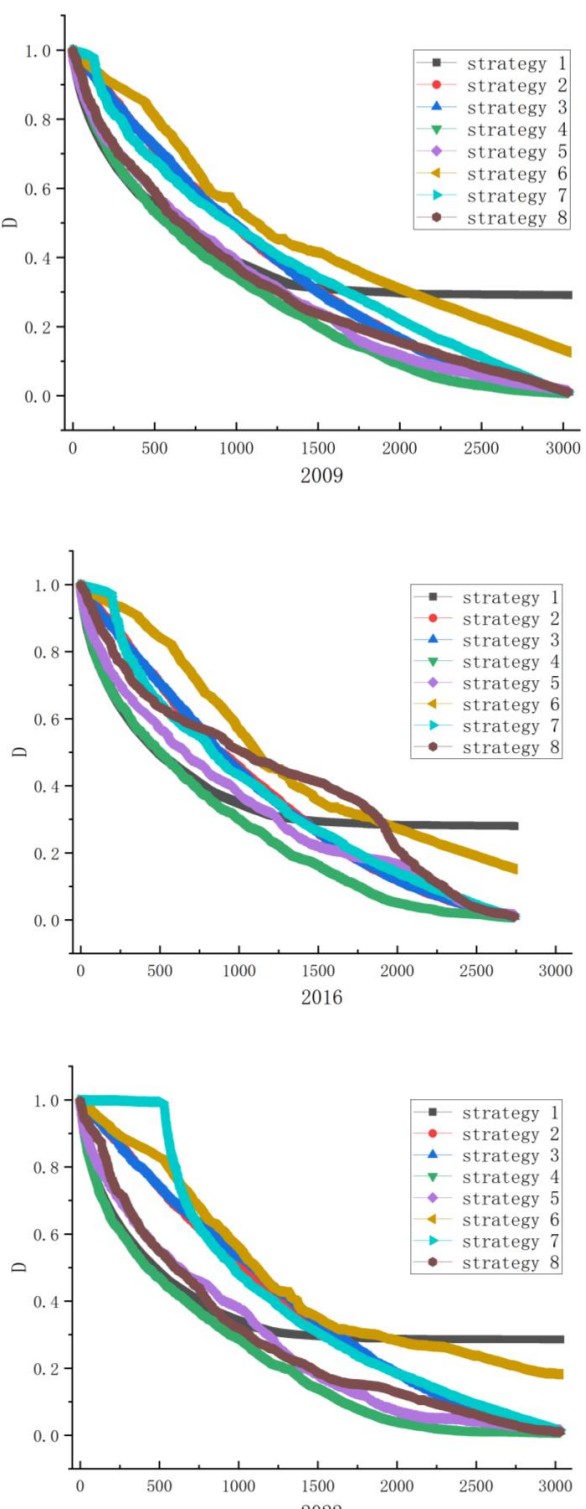

**Figure 3.** Resilience analysis based on connectivity of rail transit network.

We can find that static betweenness centrality has the fastest rate of damage to the network; it can better measure the importance of nodes and needs special attention in the process of network protection and development. The network connectivity remains best with the node clustering coefficient strategy, and the network suffers the least damage. The

random node failure and the edge failure disturbance strategy have little impact on network connectivity. The rail traffic network connectivity performance in 2009 under the entropy importance disturbance strategy and the dynamic betweenness centrality strategy has closer network connectivity, and its performance weakens in 2016 but gradually increases in 2019, while remaining consistent with the *D*-value under the dynamic betweenness centrality disturbance strategy. In the early stage of network node disturbance, the difference between the static betweenness centrality and the degree value-based disturbance strategy on network disturbance is small, and the rate of decline in network connectivity is the closest and the highest. When the network connectivity decreases to about 0.5, the curve of the degree value-based disturbance strategy flattens out and eventually maintains a relative size of 0.3, with similar performance in all years. The shortest time path summation disturbance strategy in 2022 is obviously different from that in other years, and the network connectivity increases year by year. This trend of increased resilience is most evident in 2022. When the number of disturbed nodes is 523, the size of network connectivity still remains at 0.99, which means that even the simultaneous failure of the top 523 nodes of the shortest time path sum value has little impact on the connection performance of the rail transit network. To maintain the *D*-value size at 0.8 in 2009, the top 16.9% of nodes having a higher clustering coefficient value need to be disturbed, and the top 18.19% of nodes need to fail in 2022. Maintaining the same network connectivity of 0.8 requires a smaller percentage of failed nodes under other disturbance strategies, e.g., 3.63% and 2.38% under the degree value-based disturbance strategy, respectively; 3.96% and 2.31% under the static betweenness centrality disturbance strategy; and 4.52% and 3.81% under the dynamic betweenness centrality disturbance strategy (see Figure 4).

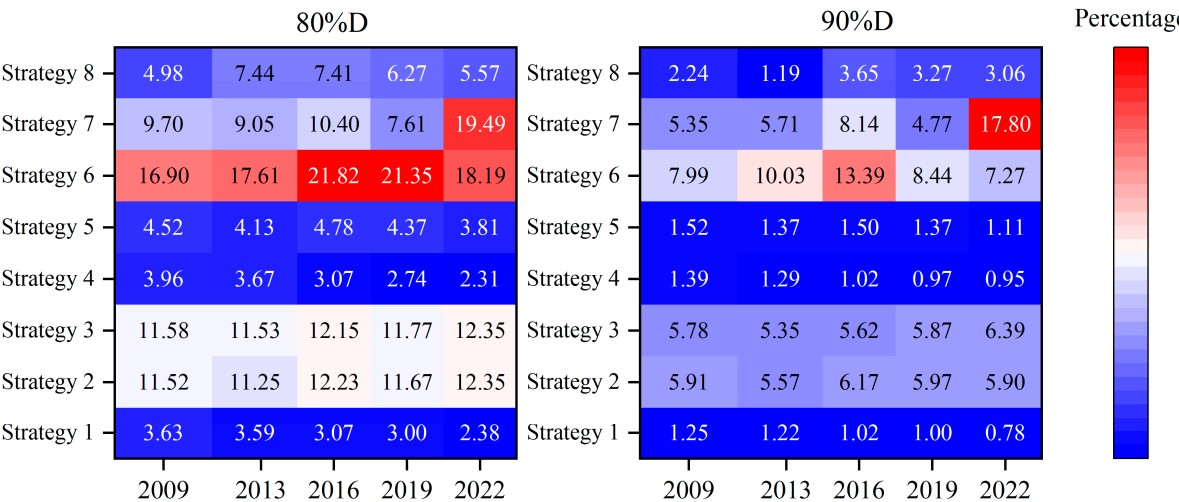

**Figure 4.** Relationship between the effectiveness of *D*-value and the failure ratio of each disturbance strategy.

Several disturbance strategies that perform well in terms of node importance for measuring network connectivity performance show that rail transit network connectivity has shown a decreasing trend year by year. This phenomenon is different from rail transit network accessibility, indicating that the resilience of rail transit is weakening year by year from the perspective of network connectivity. The country is in an important period of increasing infrastructure construction, and the rail transportation network is being continuously improved and optimized. The substantial construction of high-speed rail forces the railroad stations to gradually spread to the edge cities and counties, and it is expected that the scale of China's railroad network can reach 175,000 km by 2025. The new stations' construction and increases or decreases in rails are not completely random; they have a certain preference of dependence on important stations but still less direct contact

with other stations, with smaller degree values having less contribution to the *D*-value. The connectivity of important nodes in the network is gradually increasing.

### 3.3.2. The Resilience of RTTN Accessibility Is Increasing Year by Year

Figure 5 gives the variation in *S*-values based on the disturbances in each year, and the horizontal coordinates indicate the number of damaged network nodes. Since there are no isolated nodes in the initial state network, *S* is equal to 1. As the number of the disturbed nodes increases, the network structure is gradually damaged, and the sum of the shortest time paths between node pairs decreases. This indicates that the overall accessibility of the rail network system is weakening.

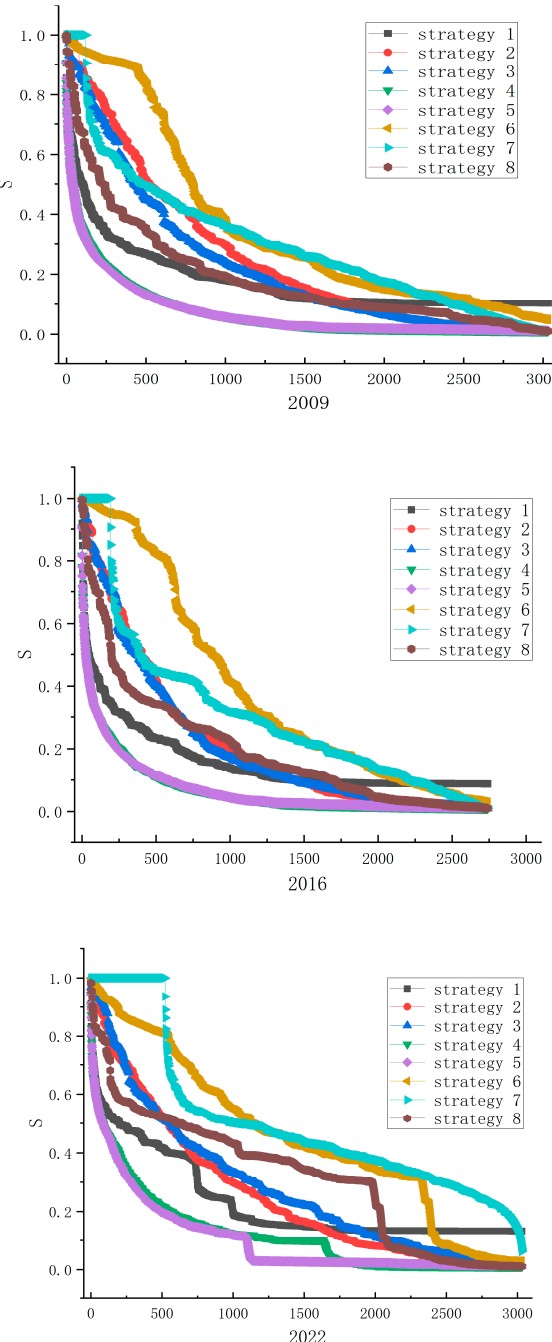

**Figure 5.** Resilience analysis based on rail network accessibility.

It can be found that the static and dynamic betweenness centrality disturbance strategy has the most obvious disturbance effect, and the network is affected with the fastest network accessibility drop rate. The failure of a single node with the largest betweenness value in 2009–2019 can cause a 10% reduction in network accessibility. In contrast, the failure of the three nodes with the largest betweenness values in 2022 can maintain only 90% of the effectiveness of network accessibility. The accessibility curves of the static and dynamic betweenness centrality disturbance strategies are highly similar in the interval of 0.9–1 for the *S*-values. In the dynamic betweenness centrality disturbance strategy, the betweenness weight value of the network changes with each failed node, and each disturbance is a destruction of the node with the largest existing network betweenness value. This differs from the static betweenness centrality disturbance strategy, which fully takes into account the consequent performance changes when the network is affected, therefore better representing the new network state and providing a better measure of the network characteristics than the static strategy. Among the two disturbance strategies mentioned above, the number of failed nodes is, in order, 40, 34, 30, 33 and 82 under the static betweenness value, whereas it is 38, 33, 27, 31 and 83 under the dynamic strategy when the network accessibility is kept 50% effective from 2009 to 2022 (see Figure 6). The percentage of failed nodes is 0.13%, 0.11%, 0.11%, 0.10% and 0.20% under the static strategy and 0.13%, 0.11%, 0.07%, 0.10% and 0.23% under the dynamic strategy when the network accessibility is kept at an 80% effective level for all years. It is easy to find that the resilience is reduced between 2009 and 2016, and the rate of accessibility failure is faster and affected more by a disturbance under the dynamic betweenness centrality disturbance strategy. In 2016–2022, the network resilience is enhanced. This is because the travel time between each rail transit station is significantly shortened: the high-speed rail passage mileage especially increases, the number of operating trips increases, passengers have more options to choose their travel routes, the damage to some nodes or trips will not hinder normal travel, and the overall network resilience is subsequently enhanced.

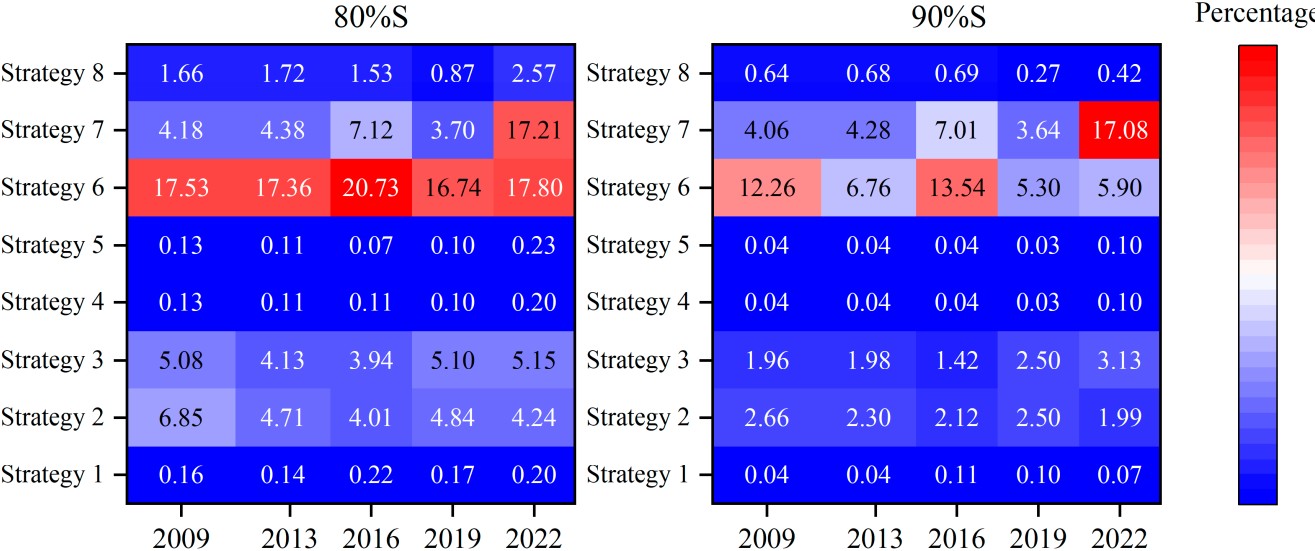

**Figure 6.** The relationship between the network accessibility of the rail network and the failure ratio of each disturbance strategy.

Under the clustering coefficient disturbance strategy, the *S*-value decreases at the smallest rate and is least affected by the disturbance, and the measure of node importance is not accurate compared with other disturbance strategies. When the sum of the shortest time paths disturbance strategy takes effect, the network accessibility does not change immediately, and the *S*-value gradually starts to decrease after the number of affected nodes exceeds 98 in 2009 and 516 in 2022. The percentage of node failure is 4.06%, 4.28%, 7.01%,

3.64% and 17.08% in each year when the *S*-value is 0.9, respectively. It can be seen that the network accessibility is significantly reduced by disturbance in 2022, and individual station failure has little impact on the normal operation of the whole network; the existing network can still meet the passengers' demand for fast travel. The node degree value-based disturbance strategy on the *S*-value has been reduced since 2016, which is related to the construction of the rail network; the newly added train stations are less connected with other stations and do not contribute much to the network stability, and thus the node failure is intensified. The random disturbance strategy of nodes and edges has an insignificant effect on network accessibility, and the entropy importance disturbance strategy curve performs normally, with a slight decrease from 2016 to 2019. In addition, the network accessibility changes in 2022 under the disturbance strategy, showing a stepwise decreasing trend.

## 4. Conclusions

The following conclusions can be drawn from the above analysis.

In China's rail transit system, the number of stations is increasing year by year, and the connection of the network is increasing while showing a small-world effect. Stations form various types of small rail clusters around different core stations, which is an effective operating network with development potential. The non-homogeneity of the system decreases, and the degree of core-periphery also decreases. From 2009 to 2022, the number of important stations increases, the inter-station travel time obviously decreases, and the accessibility level increases year by year.

The static node betweenness centrality-based disturbance strategy has the fastest damage rate and the most obvious effect on the connectivity performance of the rail network. The network has the strongest resilience ability with the sum of the shortest time path disturbance strategy, especially in the pre-disturbance period, and this is particularly evident in 2022. In addition, the degree value and static betweenness centrality disturbance strategies are close to the damage to the network connectivity when the network suffers from a small number of important damaged stations. The entropy importance disturbance strategy has a similar damage ability to the dynamic betweenness centrality disturbance strategy. There is a trend of general decline in rail network connectivity performance, which is due to weaker connectivity between additional stations and surrounding stations, continued enhancement of connectivity at important stations, and successive failures of important stations, resulting in a faster rate of decline in overall network performance and weakened resilience.

Static and dynamic betweenness centrality disturbance strategies have the fastest rate of decline in rail network accessibility. The dynamic disturbance strategy sees more obvious damage to the network structure, and more attention should be paid to protect nodes with higher values of dynamic betweenness centrality, accordingly. The network accessibility is weakened from 2009–2016, and then gradually increases from 2016–2022. The proportion of high-speed trains in rail transit increases, the inter-station travel time decreases significantly, and the impact of individual station failure on the overall accessibility gradually decreases.

In the measurement of both connectivity and accessibility of the rail network, the betweenness centrality strategy performs optimally, so that when the network is damaged, the stations can be repaired in order according to the betweenness centrality value. This can speed up the restoration of train operations between important stations and ensure the maximum recovery rate of the network structure. In maintenance construction focused on important stations or hub stations, attention should also be paid to the interconnection between other stations to enhance the resilience of the network structure.

Based on the above foundation, our future research can be carried out in the following aspects: first, we can combine railroad operation timetable data and railroad passenger flow data to evaluate the changes in the resilience of a railroad transportation network under different conditions and to perfect the network recovery strategy. Second, we can consider adding the data of highway network and road network to optimize the passenger transfer

scheme and enrich travel options so as to enhance the stability of the road transport network. Alternately, we can construct relevant models to refine the emergency mechanisms of the railroad transportation network and improve the network operation efficiency.

**Author Contributions:** Conceptualization, R.D.; methodology, R.D. and L.D.; validation, L.D., Y.D. and J.F.; formal analysis, L.D. and J.F.; investigation, L.D.; resources, Y.D., Y.Z. (Yuqi Zhu) and L.P.; data curation, Y.Z. (Yilin Zhang); writing—original draft preparation, L.D. and J.F.; writing—review and editing, R.D.; visualization, R.D. and L.D. All authors have read and agreed to the published version of the manuscript.

**Funding:** This work was supported by the National Natural Science Foundation of China (No. 72001053).

**Institutional Review Board Statement:** Not applicable.

**Informed Consent Statement:** Not applicable.

**Data Availability Statement:** The data presented in this study are available on request from the corresponding author.

**Conflicts of Interest:** The authors declare no conflict of interest.

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
