# Peer review of "Study on the Evolution and Resilience of Rail Transit Time Networks—Evidence from China"

_applsci, doi:10.3390/app12199950_

Round 1

Reviewer 1 Report

The work presented in this research is appreciable. Kindly remodify the abstract and replace words such as "betweenness centrality" with other similar words. Moreover the absract may contain the results from the conclusion part, you may add.

Author Response

The work presented in this research is appreciable. Kindly remodify the abstract and replace words such as "betweenness centrality" with other similar words. Moreover, the abstract may contain the results from the conclusion part, you may add.

Response:

Thank you very much for your great comments and suggestions. Regarding the revision of the abstract you mentioned, we have added the results and some numerical expressions. The revised abstract is shown as: “In the network operation and management of rail transit systems, the occurrence of unexpected events causes damage to the network structure, further hindering the regional accessibility performance and the function of the system. This study is based on the rail transit operation schedules in 2009, 2013, 2016, 2019 and 2022, we construct a directional weighted rail transit time network (RNNT) with train operation time as the weight, compared the betweenness centrality, sum of shortest time path and entropy importance etc., and quantitatively measure the network accessibility, connectivity and its resilience evolution. The results show that the current rail transportation network in China has a "small-world" effect and there are a few stations with strong connections. The most densely distributed intervals of travel times between pairs of nodes changes from [440,445] to [207,210]. The fastest and best-performing disturbance to network connectivity and accessibility performance are both caused by the betweenness disturbance strategy. When the network connectivity remains 80% effective, the ratio of failed nodes under static betweenness centrality strategy decreases from 3.96% in 2009 to 2.31% in 2022, with weaker connections between node pairs and their network resilience diminishes. When the network accessibility remains 80% effective, the ratio of failed nodes under static (dynamic) betweenness centrality strategy increases from 0.13% (0.13%) in 2009 to 0.20% (0.23%) in 2022. Therefore, the rail transit network can protect the corresponding rail stations based on the station ranking of the above strategies, and this research is beneficial to rail transit network protection and structure optimization.”

The concept of "betweenness centrality" was introduced by American sociologist Professor Lyndon Freeman (1979). In this study, this concept was applied as one of the indicators to measure the importance of each rail station in each year, and therefore, after careful consideration, "betweenness centrality" is still used here.

Reviewer 2 Report

Dear Authors,

I would like to thank you for the opportunity to read your manuscript entitled “Study on the evolution and resilience of rail transit time networks – evidence from China”.

The research is interesting and topical as it addresses the issue of the network operation and management of rail transit systems in case of relevant events.

The paper is structured and organized as follows: there is a first section with the introduction and overview of the work, then a section with Network Structure Analysis, a section where the research results are presented, and finally, the conclusions.

All figures and tables are correctly cited in the text.

Below are my major revisions hoping that they will be helpful to you in improving and enhancing the paper:

1.     Section 1 (Introduction and Overview) seems a bit confusing to me. I would suggest keeping a first section by which you introduce the work, also highlighting the motivations, and adding a second section referring to the scientific-technical state of the art. This makes the starting point of your work more understandable, considering the international research panorama.

2.     At the end of section 1, I would include a description of the organization and structure of the paper. This will help the reader to read the paper.

3.     In the conclusion section, I would highlight possible developments in the work.

4.     Finally, with reference to the topic of resilience, of managing railroad nodes in emergencies, I suggest you look at the paper where you can take some ideas:

·       Borghetti, F., & Malavasi, G. (2016). Road accessibility model to the rail network in emergency conditions. Journal of Rail Transport Planning and Management, 6(3), 237-254. doi:10.1016/j.jrtpm.2016.10.001

https://www.scopus.com/record/display.uri?eid=2-s2.0-84994560256&origin=resultslist&sort=plf-f

Thank you and good luck.

Author Response

  • Section 1 (Introduction and Overview) seems a bit confusing to me. I would suggest keeping a first section by which you introduce the work, also highlighting the motivations, and adding a second section referring to the scientific-technical state of the art. This makes the starting point of your work more understandable, considering the international research panorama.

Response:

Thank you very much for your detailed suggestions. We have adjusted and focused on revising parts of the introduction and the literature review. Firstly, we start from the world railroad development, highlight the role of railroads and their importance in the rail transportation network, and finally put the research perspective on China, highlighting the Chinese government issued corresponding national policies and pay very high attention to the construction of rail transportation network.

As “The United States has a long history of railway development, 80% of which is freight transportation. With the development of geo-economic changes, many countries in the world attach great importance to the important role of railroads in strengthening regional economic, political, social and cultural ties and safeguarding national security, and regional road networks tend to be integrated. In the North American Free Trade Area, the United States, Canada and Mexico are closely linked by railway, and in order to strengthen the connection of railroad routes in each country, the construction of the pan-European railroad network has been gradually enhanced, and the construction of the pan-Asian railroad network is also being actively promoted. While Japan Shinkansen is recognized as one of the safest high-speed railways in the world, and its operation safety management is at the international leading level. European railway lines have been extended to domestic countries to form a relatively stable network, many urban rail transit networks have been extended to all directions of the city. The "Outline of the Construction of a Strong Transportation Country" and "Outline of the National Comprehensive Three-dimensional Transportation Network Planning" issued by the Central Committee of the Communist Party of China (CPC) and the State Council have clearly proposed to build an integrated urban transportation network and enhance the resilience of China's rail transportation system in the future [1-2].”

The motivation of this paper is mainly on how to properly analyze the network performance and stabilize the network structure so as to enhance the network resilience. As “Existing research argues that resilience can integrate the resilience and recovery capacity of transportation systems in facing external shocks [14], and its formation and evolution process will help us, to understand the performance change patterns of rail transportation networks under environmental or man-made disturbance, so that managers can clarify network resilience identification, metrics and optimization methods, which is beneficial to the operation and emergency management of rail transportation network systems. In an environment where urban connectivity is increasingly diverse and transportation network structure is more complex, there are still shortcomings in the resilience of the rail network, and the ability of the network to cope with interference and resist attacks needs to be enhanced.”

Based on complex networks, this study analyzes the structure and performance of rail transportation networks and introduces topological indicators to evaluate the connectivity and accessibility of Chinese rail transportation networks with reference to existing literature. Scholars have paid less attention to railroad transportation time, railway network has long-term and difficult to change characteristics, railroad train time and train grouping is more fixed, so using inter-station running time as time weight to construct a directed network to analyze network resilience and recovery ability is more in line with the realistic needs.

  • At the end of section 1, I would include a description of the organization and structure of the paper. This will help the reader to read the paper.

Response:

Thank you very much for your detailed suggestion on this section, which is one of the elements missing in this study. Therefore, we have reorganized the existing statements at the end of the first part and added an analysis of the organizational structure of the paper, listing the logic of the lines in steps.

The details are as follows: “Therefore, in this study, based on the railroad operation schedule for a total of five years from 2009 to 2022, a directional time-weighted network model is constructed on the basis of a complex network. Then analyze the characteristics of the network such as small-world property, connectivity and accessibility. Further propose node importance evaluation indexes based on degree, betweenness centrality, and the sum of the shortest paths etc. Finally, quantitatively compare the stable state of the network structure under different disturbance strategies and analyze the network connectivity and accessibility. So as to measure the trend of network resilience evolution, in order to provide a basis for the railway transport system management to develop emergency response plans.”

  • In the conclusion section, I would highlight possible developments in the work.

Response:

Thank you very much for your suggestion. Based on the existing research content, we have considered in detail the next research directions and proposed several plans with practical and research significance, and sincerely look forward to the early implementation of these research plans. As “Based on the above foundation, our future research can be carried out in the following aspects: firstly, we can combine railroad operation timetable data and railroad passenger flow data, to evaluate the changes in the resilience of railroad transportation network under different situations, to perfect the network recovery strategy. Secondly, we can consider adding the data of highway network and road network to optimize the passenger transfer scheme and enrich the travel options, so as to enhance the stability of road transport network. Or construct relevant models to refine the emergency mechanism of the railroad transportation network and improve the network operation efficiency.”

  • Finally, with reference to the topic of resilience, of managing railroad nodes in emergencies, I suggest you look at the paper where you can take some ideas:

Response:

We sincerely thank you for your detailed suggestions on this study and also for recommending us the very informative literature, which we have read in detail and benefited from. Benefiting from the relevant content of the article, we have used it as reference [11] as a strong support for our research content.

Reviewer 3 Report

1) It is suggested to shorten the abstract. Only the main technical flow and main finding of significance should be presented. Some quantitative results should be shown to support the main conclusions.

2) In the introduction, it is recommended to include one sentence to introduce that the source of the failure of the rail network is mainly from the infrastructure [*1] and vehicle issues [*2].

[*1] Song, Yang, et al. "Wind deflection analysis of railway catenary under crosswind based on nonlinear finite element model and wind tunnel test." Mechanism and Machine Theory 168 (2022): 104608.

[*2] Wei, Xiukun, et al. "On fault isolation for rail vehicle suspension systems." Vehicle System Dynamics 52.6 (2014): 847-873.

3) In the introduction, it is not clear the definition of the railroad network. Does it only contain the electric railway?

4) It is not clear why the data in 2009, 2013, 2016… are collected. 4 years gap between 2009 and 2013, but 3 years gap between others.

5) Please add references to ‘with references to scholars’ research complex networks’.

6) Figure legend should be included in figure 1. It is not straightforward to see the meaning of these nodes.

7) The legends in figure 3 should be revised. The line shapes cannot be seen.

8) In line 305, it is desirable to further explain the small-world effect.

Author Response

1) It is suggested to shorten the abstract. Only the main technical flow and main finding of significance should be presented. Some quantitative results should be shown to support the main conclusions.

Response:

Thank you very much for your detailed and specific revisions in the abstract section. We have also noticed that the abstract is a bit long and lacks quantitative numerical results. Therefore, we have reorganized the abstract to highlight the main research steps and the extremely important conclusions, based on which we have tried to streamline the language presentation and shorten the length of the abstract. More precise modifications are shown below.

“ This study is based on the rail transit operation schedules in 2009, 2013, 2016, 2019 and 2022, we construct a directional weighted rail transit time network (RNNT) with train operation time as the weight, compared the betweenness centrality, sum of shortest time path and entropy importance etc., and quantitatively measure the network accessibility, connectivity and its resilience evolution. The results show that the current rail transportation network in China has a "small-world" effect and there are a few stations with strong connections. The most densely distributed intervals of travel times between pairs of nodes changes from [440,445] to [207,210]. The fastest and best-performing disturbance to network connectivity and accessibility performance are both caused by the betweenness disturbance strategy. When the network connectivity remains 80% effective, the ratio of failed nodes under static betweenness centrality strategy decreases from 3.96% in 2009 to 2.31% in 2022, with weaker connections between node pairs and their network resilience diminishes. When the network accessibility remains 80% effective, the ratio of failed nodes under static (dynamic) betweenness centrality strategy increases from 0.13% (0.13%) in 2009 to 0.20% (0.23%) in 2022. Therefore, the rail transit network can protect the corresponding rail stations based on the station ranking of the above strategies, and this research is beneficial to rail transit network protection and structure optimization.”

2) In the introduction, it is recommended to include one sentence to introduce that the source of the failure of the rail network is mainly from the infrastructure [*1] and vehicle issues [*2].

Response:

Thank you very much for such a detailed suggestion for improvement. After reading these two articles you recommended, we think that the sentence you listed is very valuable to us and plays a great, top-to-bottom role. Therefore, we have added these two articles as references [5] and [6] to our description of the railroad network to make our presentation more convincing.

3) In the introduction, it is not clear the definition of the railroad network. Does it only contain the electric railway?

Response:

Thank you for your suggestion, and with your reminder, we note that this section does lack specific description. In this study, the railroad network is defined as an integrated transportation network of electric railway and non-electric railway. To address this issue, we have revised the last paragraph of introduction part and added the definition of the railroad network as "The railroad network studied in this article is defined as an integrated transportation network of electric railway and non-electric railway, and our study is mainly based on the rail transit routes on these integrated railroad network."

4) It is not clear why the data in 2009, 2013, 2016… are collected. 4 years gap between 2009 and 2013, but 3 years gap between others.

Response:

Thank you very much for your question. These dates are determined based on when we crawled the corresponding data in different years. Due to the system update and other problems, it is impossible to climb the corresponding historical data now, so we can only choose to use the already have historical data and collect the latest data of 2022. At the same time, the time period should be kept as long as three years as possible, so that the changes of the network can be more prominent, but only the gap between 2009 and 2013 is four years (if we choose 2021, then the data will not be the newest one).

5) Please add references to ‘with references to scholars’ research complex networks’.

Response:

Thank you for your suggestion. During the revision process, some literatures have been added to this section as references [26-28] to corroborate our study description.

6) Figure legend should be included in figure 1. It is not straightforward to see the meaning of these nodes.

Response:

Thank you very much for your suggestion. The content of Figure 1 is indeed not very easy for the reader to understand, so we have presented the actual analysis results as a legend in Figure 1 and adjusted the clarity of the figure, hoping that the graphical presentation now will help you better understand the content of our study.

7) The legends in figure 3 should be revised. The line shapes cannot be seen.

Response:

Thank you very much for your suggestion. We have listened carefully to your suggestions and tried to modify the lines in Figure 3. However, due to the specificity of the actual results, the attack strategies 2 and 3 are very close to each other in terms of the degree of damage to the network performance, and the other attack strategies are also closer to each other in terms of the way they damage the network, so the line curves overlap. We modified the width of the lines, the presentation, etc., but in the end, we could not clearly distinguish the lines for each attack strategy. Your suggestion is very pertinent and clear, and we are sorry that we have been unable to improve it through various methods, so please understand.

8) In line 305, it is desirable to further explain the small-world effect.

Response:

Thank you for your suggestion. We have noted the lack of conceptual representation of the "small-world" effect in the paper, with your suggestion we have added the explanation of small-world effect as ”The small-world model was proposed and introduced by Watts and Strogatz in 1998 to the study of complex networks, which can be determined in terms of both higher agglomeration coefficient and shorter average path length. Most networks in real social activities have a small-world effect. Which means that most nodes are only closely connected to their neighbors, it can explain the emergence of multiple network forms and facilitate the establishment of close cooperation within the network, and reach any other node in the network by passing through only a few nodes, random reconnections occur between some pairs of distant nodes, which build bridges between groups across small groups with strong internal relationships.”

Round 2

Reviewer 2 Report

Dear Authors,

thank you for considering my suggestions. I hope my recommendations have helped to improve your research.

I have seen that you have responded in a timely and comprehensive manner to the 4 requests for further study: 1) Introduction and Overview, 2) Structure of the paper, 3) Conclusions and possible developments and 4) reference.

Therefore, I believe that the research is publishable without further modification.

Thank you and good luck.